# Data-Driven and Machine Learning to Screen Metal–Organic Frameworks for the Efficient Separation of Methane

**DOI:** 10.3390/nano14131074

**Published:** 2024-06-24

**Authors:** Yafang Guan, Xiaoshan Huang, Fangyi Xu, Wenfei Wang, Huilin Li, Lingtao Gong, Yue Zhao, Shuya Guo, Hong Liang, Zhiwei Qiao

**Affiliations:** 1Guangzhou Key Laboratory for New Energy and Green Catalysis, School of Chemistry and Chemical Engineering, Guangzhou University, Guangzhou 510006, China; 2005200075@e.gzhu.edu.cn (Y.G.); xshuang@e.gzhu.edu.cn (X.H.); 32105200048@e.gzhu.edu.cn (F.X.); 2112105103@e.gzhu.edu.cn (W.W.); lihuilin1@e.gzhu.edu.cn (H.L.); 32305200102@e.gzhu.edu.cn (L.G.); 2State Key Laboratory of NBC Protection for Civilian, Beijing 100191, China; sa11226532@mail.ustc.edu.cn

**Keywords:** methane, metal–organic frameworks, gas separation, molecular simulation, machine learning, diffusion

## Abstract

With the rapid growth of the economy, people are increasingly reliant on energy sources. However, in recent years, the energy crisis has gradually intensified. As a clean energy source, methane has garnered widespread attention for its development and utilization. This study employed both large-scale computational screening and machine learning to investigate the adsorption and diffusion properties of thousands of metal–organic frameworks (MOFs) in six gas binary mixtures of CH_4_ (H_2_/CH_4_, N_2_/CH_4_, O_2_/CH_4_, CO_2_/CH_4_, H_2_S/CH_4_, He/CH_4_) for methane purification. Firstly, a univariate analysis was conducted to discuss the relationships between the performance indicators of adsorbents and their characteristic descriptors. Subsequently, four machine learning methods were utilized to predict the diffusivity/selectivity of gas, with the light gradient boosting machine (LGBM) algorithm emerging as the optimal one, yielding *R*^2^ values of 0.954 for the diffusivity and 0.931 for the selectivity. Furthermore, the LGBM algorithm was combined with the SHapley Additive exPlanation (SHAP) technique to quantitatively analyze the relative importance of each MOF descriptor, revealing that the pore limiting diameter (PLD) was the most critical structural descriptor affecting molecular diffusivity. Finally, for each system of CH_4_ mixture, three high-performance MOFs were identified, and the commonalities among high-performance MOFs were analyzed, leading to the proposals of three design principles involving changes only to the metal centers, organic linkers, or topological structures. Thus, this work reveals microscopic insights into the separation mechanisms of CH_4_ from different binary mixtures in MOFs.

## 1. Introduction

Coal mine methane (CMM) [1] is methane released during the coal mining process, typically mixed with gases such as nitrogen, oxygen, and carbon dioxide. CMM is primarily stored in the coal matrix through physical adsorption, where the coal matrix contains a large number of micropores and mesopores; the high specific surface area of micropores makes them crucial for methane adsorption [2]. In abandoned coal mines, microbial methane production is a new source of methane, indicating that, even after coal mining ceases, methane generation and emission remains a concern that needs attention [3]. Accurately predicting the emission of coal mine methane [4] is essential for understanding its potential environmental impact and planning the use of methane as an energy source. Methane in nature often contains impurities, such as nitrogen, carbon dioxide, and hydrogen sulfide, which can reduce the combustion efficiency of methane, corrode pipelines, and increase the environmental burden. Wu [5] and others developed a new amine ion-exchanged zeolite Y by introducing amine cations CH^+^ and TWA^+^ to the original zeolite Y and confirmed that they had good separation performance in the CH_4_/N_2_ system. The global warming potential of each mole of methane is 3.7 times that of carbon dioxide, and carbon dioxide emissions account for 80% of the contribution of current greenhouse gas emissions to global warming [6]. Although methane is a greenhouse gas, through effective capture and utilization, methane can also serve as an efficient and clean energy source, playing a positive role in global energy transition and climate change mitigation [7]. Biogas, formed by mixing methane with an appropriate amount of air, is an ideal gas fuel. Purifying biogas for use as vehicle fuel can further improve the utilization rate of biogas [8]. The pressure swing adsorption (PSA) purification method utilizes the selective adsorption of carbon dioxide by the adsorbent [9], that is, carbon dioxide has a higher separation coefficient relative to other gaseous components on the adsorbent, to achieve the purpose of removing carbon dioxide from biogas and purifying methane [10]. Purifying methane in biogas using physical or chemical methods, such as pressure swing adsorption (PSA) or chemical absorption, usually involves repeated pressurization and depressurization processes, which can lead to high energy consumption and production costs, affecting the economic benefits and sustainable development of enterprises [11].

In recent years, a new type of adsorbent material, metal–organic framework (MOF), has emerged as a novel class of porous material. Formed through self-assembling metal ions or clusters with organic ligands, MOFs boast high porosity, large surface area, and tunable pore sizes. They have been extensively studied and applied in various fields, including gas adsorption and separation [12], storage [13], catalysis [14], drug delivery [15], and sensing applications [16]. The structure of MOFs can be regulated by selecting different metal ions and organic ligands, thereby achieving precise control over pore size, shape, and chemical environment [17]. Understanding multi-component adsorption equilibria on MOFs is essential for designing adsorption-based separation processes [18]. Although a large number of MOFs have been synthesized and reported to date, given the vast variety of metal ions and organic linkers and diverse connection sequences, manually sifting through a vast database to select MOFs for a specific system using only experimental methods is undoubtedly inefficient.

Therefore, molecular simulation, high-throughput computing combined with machine learning (ML), and molecular fingerprints (MFs) are used to replace traditional research methods in searching for and designing suitable new MOF absorbents [19]. Anbia [20] and others studied the adsorption characteristics of MOF-235 for CH_4_, H_2_, and CO_2_ by measuring volume, finding that the absolute amount of adsorption was in the order of CH_4_ > H_2_ > CO_2_. The selectivity of MOF-235 for CH_4_ was higher than that for CO_2_ (14.7) and H_2_ (8.3), indicating that MOF-235 is a potential adsorbent for CH_4_ separation in gas mixtures. Niu [21] and others reported a methane nano-trap, discovering that an alkyl-MOF-based methane nano-trap has a high methane adsorption rate and CH_4_/N_2_ selectivity at 298 K and 1 bar. Yang [22] and others synthesized a microporous hetero metal–organic framework (MOF) material named CuIn(Ina)_4_, which showed excellent adsorption performance in separating and purifying CH_4_ from a ternary mixed system containing C_2_H_2_ and CO_2_. Mulu [23] and others determined that original wood ash, after calcination treatment, can effectively adsorb and remove CO_2_ and H_2_S from biogas, thereby increasing the purity of methane in biogas. Wang [24] and others designed and synthesized a series of porous coordination polymers (PCPs) with hourglass-shaped nanochannels, one of which, the porous coordination polymer NTU-30, showed excellent separation efficiency in C_2_H_6_/CH_4_ and C_2_H_4_/CH_4_ gas systems. Experiments proved that NTU-30 has weaker adsorption for CH_4_, but stronger adsorption capacity for C_2_H_6_ and C_2_H_4_, making it an adsorbent material for natural gas purification and increasing CH_4_ purity. Yang [25] and others studied the field of low-concentration coal mine methane and used the VPSA (vacuum pressure swing adsorption) method with modified carbon molecular sieves as adsorbents for methane adsorption. The results showed that this adsorption technology is particularly suitable for purifying raw gases containing low concentrations of methane. Tu [26] and others reported a nickel-based metal–organic framework Ni-BPZ, which can effectively achieve methane separation and purification, including capturing methane in coal mine gas, and revealed its efficient separation mechanism through molecular simulation. Kang [27] found in his research that, compared with other gas molecules, the MOFJUC-150 membrane has a significant preference for H_2_ permeation, and at room temperature, the selectivity factor of this membrane for H_2_/CH_4_ is 26.3.

The purpose of this work is to employ large-scale computational screening and ML to study the adsorption and diffusion properties of 6013 MOF adsorbents in six gas mixtures (H_2_/CH_4_, N_2_/CH_4_, O_2_/CH_4_, CO_2_/CH_4_, H_2_S/CH_4_, He/CH_4_). In Section 2, the atomic models of CoRE-MOF and gases, simulation methods, and principles of machine learning are described. In Section 3, the relationship between the performance indicators of the adsorbents and their characteristic descriptors is first discussed through univariate analysis, and then the best ML model for a specific performance indicator for the six systems is identified using four ML techniques, followed by the application of SHAP for a quantitative analysis of the relative importance of each CoRE-MOF descriptor. Finally, high-performance MOF materials for different systems are screened, and the commonalities among high-performance MOFs are analyzed to propose design principles.

## 2. Model and Methods

### 2.1. Molecular Model

In this work, Chung [28] and colleagues conducted a series of experiments to screen CoRE-MOFs by removing free solvent molecules using high-throughput molecular simulation and established a database containing MOF crystal models. Five structural descriptors of MOF were calculated, including pore-limiting diameter (PLD (Å)), large cavity diameter (LCD (Å)), volumetric surface area (VSA (m^2^/cm^3^)), void fraction (*ϕ*), and MOF density (*ρ* (kg/m^3^)). The LCD and PLD were calculated using zeo++ software (version 0.3) [29]. The VSA and *ϕ* were computed using the RASPA software (version 1.9.15) [30] with He with a diameter of 2.58 Å and N_2_ with a diameter of 3.64 Å as probes, respectively. In the grand canonical Monte Carlo (GCMC) simulations, the Lennard-Jones (LJ) potential function was used to describe the interactions between atoms in CoRE-MOFs, allowing particles to freely enter and exit the simulation system, thereby simulating the adsorption behavior of the system under different chemical potentials.
(1)$$u_{LJ+elec}\left(r\right)=\sum 4\varepsilon_{ij}\left[{\left(\frac{\sigma_{ij}}{r_{ij}}\right)}^{12}-{\left(\frac{\sigma_{ij}}{r_{ij}}\right)}^{6}\right]+\sum \frac{q_{i}q_{j}}{{4\pi \varepsilon }_{0}r_{ij}}$$

In Equation (1), *r_ij_* represents the distance between two atoms; *q_i_*, *q_j_* denote the charge quantities of the two atoms, respectively; *ε*_0_ = 8.8542 × 10^−12^ C^2^ N^−1^m^−2^ is the vacuum permittivity, which is used to describe the Coulombic interactions between atoms. The Lennard-Jones (LJ) potential parameters for MOFs are derived from the universal force field (UFF) [31] and are listed in Appendix A.

We constructed a database for seven gas components (CH_4_, N_2_, H_2_S, O_2_, CO_2_, H_2_, He). The force field parameters for these seven gas components listed in Appendix A are derived from the transferable potential of the Trappe force field [32]. CH_4_ is a united atom model; N_2_ is a three-site model with an N-N bond length of 1.10 Å; H_2_S is a four-site model with an S-H bond length of 1.13 Å, featuring L-J potentials on the S and H atoms. Additionally, there is a dummy atom near the S atom, with partial charges on the H atoms and the dummy atom, while the S atom is uncharged. O_2_ is a three-site model, and for CO_2_, the C-O bond length is 1.16 Å and an ∠OCO was 180°. The He atom has a diameter of 2.58 Å.

### 2.2. Molecular Simulations

In this study, the temperature was set to 298 K and the pressure to 1 bar. The diffusion coefficients *D* of CH_4_, N_2_, H_2_S, O_2_, CO_2_, H_2_, and He in the MOF were calculated using MD simulation [33]. Each MOF underwent an independent MD simulation to individually assess its diffusion characteristics for gas molecules. The time step for the MD simulation was 1 fs, which was the smallest time interval used in the simulation to calculate atomic motion. The duration of each MOF’s MD was 5 ns, with the first 3 ns used for the system to reach an equilibrium state, and the last 2 ns for production. The data from the production phase will be used for the final analysis and results. The diffusion coefficient in the MD simulation was calculated from the mean square displacement using the Einstein equation. All MD simulations were performed using the RASPA software package. The simulation using this method showed better consistency with experiments (see Appendix A for details). To quantify the diffusion separation of two gases, the diffusion selectivity between ideal binary gas pairs was estimated using Equation (2).
(2)$$S_{diff(i/j)}=\frac{D_{i}}{D_{j}}$$

In this context, *D_i_* and *D_j_* represent the diffusion coefficients of gas molecules i or j within CoRE-MOFs, where i stands for N_2_, H_2_S, O_2_, CO_2_, H_2_, He, and j stands for CH_4_. The pairs *i*/*j* represent different gas combinations, listed in Appendix A.

The interactions between CoRE-MOFs and adsorbate molecules were calculated using the Lorentz–Berthelot rules. Periodic boundary conditions (PBC) are a standard simulation technique, enabling the representation of an infinitely large material within a small repeating unit. In a 3D system with PBC, particles that exit one side of the simulation box re-enter from the opposite side, ensuring continuity. The simulation cell, which contains the particles, is expanded to a minimum of 24 Å in every dimension to ensure accurate modeling. To calculate the Lennard-Jones (LJ) interactions, a long-range correction spherical cutoff radius of 12 Å was set.

### 2.3. Machine Learning

In this work, four algorithms were used to predict two target values—diffusion coefficient (*D*) and diffusion selectivity (*S_diff_*). In machine learning, constructing a diverse training library is crucial as it helps ensure the model has generalization capabilities, that is, it can accurately predict new unseen data. This training library includes gas molecules of different sizes, shapes, and polarities (CH_4_, N_2_, H_2_S, O_2_, CO_2_, H_2_, He), and CoRE-MOFs with different topological structures and chemical compositions. A total of 6013 MOFs with different topological structures and chemical compositions were sourced from the 2019 CoRE-MOF database. Additionally, to facilitate the exploration of diffusion similarities of different molecules within MOFs, the dataset was arranged longitudinally.

Firstly, molecular dynamics simulations (MD) were used to obtain the diffusion coefficients (*D*) of seven gas molecules (CH_4_, N_2_, H_2_S, O_2_, CO_2_, H_2_, He), resulting in 42,092 preprocessed data entries. The feature variables consisted of MOF structural descriptors (PLD, LCD, VSA, *ϕ*, *ρ*) and gas physical properties (see attachment Appendix A) (kinetic diameter (*Dia*), quadrupole moment (*Qua*), dipole moment (*Dip*), and polarizability (*Pol*)). The diffusion selectivity (*S_diff_*) for six ideal binary gas mixtures was derived using Equation (2), resulting in 36,078 preprocessed data entries. The feature variables consisted of MOF’s five structural descriptors (PLD, LCD, VSA, *ϕ*, *ρ*), the physical properties of the separated gases (kinetic diameter (*Dia*), quadrupole moment (*Qua*_i_), dipole moment (*Dip*_i_), polarizability (*Pol*_i_)), and differences in the physical properties of the separated gases (Δ*Dia*, Δ*Pol*, Δ*Dip*, Δ*Qua*) [34]. The data then proceeded to the next step of machine learning.

To explore the importance of geometric/energy descriptors (LCD, PLD, VSA, *ρ*, *ϕ*) in MOFs for their performance, four machine learning (ML) algorithms were used for data mining, including random forest (RF) (version 1.2.2) [35], gradient boosting regression trees (GBRT) in scikit-learn (version 1.0.2) [36], extreme gradient boosting (XGB) (version 1.1.2) [37], and lightGBM (LGBM) (version 3.3.2) [38] for comparison. All algorithms were executed in scikit-learn for Python [39], and detailed information on the principles of ML algorithms can be found in Supporting Information Appendix A. All version numbers of the Python packages used during training are listed in Appendix A. ML algorithm parameters can be found in detail in Appendix A. Before training the ML models, the dataset was first randomly divided into training and testing sets at a ratio of 7:3, followed by standardization processing. During training, k-fold cross-validation was used (k = 5 in this work) to improve the stability and accuracy of the model; the principle can be found in Appendix A. In machine learning and statistical modeling, evaluating the performance of the model is a very important step. The coefficient of determination *R*^2^, mean absolute error (MAE), and root mean square error (RMSE) are three commonly used evaluation metrics. A higher *R*^2^ indicates greater predictive accuracy, while lower MAE and RMSE values signify enhanced stability in the model’s predictions.

In this work, SHapley Additive exPlanation (SHAP) technology was combined with machine learning models for the analysis of the relative importance of features. SHAP (version 0.40.4) [40] is a popular interpretive tool for machine learning that helps us understand the specific impact of each feature of the model on the model’s predictions. It can be calculated through the TreeExplainer algorithm developed by Lundberg [41] and others. TreeExplainer is a method for calculating SHAP values to interpret tree-based models. For each sample in the dataset, TreeExplainer individually calculates the result of that sample in each decision tree of the model. It then assesses the contribution of each feature to the sample’s predicted outcome in each decision tree. By analyzing the SHAP values across all samples, the global behavior of the model can be interpreted.

## 3. Results and Discussion

### 3.1. Statistical Analysis

To explore the impact of MOF structure on the diffusion performance of seven gases during the diffusion process, we first analyzed the relationship between diffusivity (*D*) and five descriptors. As shown in Figure 1a–d, *D* increases with the increase in PLD, LCD, *ϕ*, and VSA, eventually tending to stabilize, as they are all descriptors of pore size or pore volume. *ρ* is negatively correlated with diffusion performance, showing a downward trend as *ρ* increases, which may be because *ρ* is not only related to the material’s pore channels but also to its mass. When *ρ* > 3000 kg/m^3^, the dispersion of *D* is very high, as shown in Figure 1e. MOFs typically have a high specific surface area and adjustable pore structure, which makes them excellent for gas storage (such as hydrogen, methane, and carbon dioxide). However, *ρ* may vary greatly due to the type and quantity of the constituent elements and may include heavier metal atoms such as gold (Au), platinum (Pt), or uranium (U), which can significantly increase the overall density of the structure. Although some MOFs may have a larger free space (high porosity), which can provide more gas storage capacity, the density of MOFs may still be high. Therefore, *ρ* is not easy to represent the performance of MOFs.

To further analyze, the relationship between *D* and PLD for the seven gases (CH_4_, N_2_, H_2_S, O_2_, CO_2_, H_2_, He) was explored. The kinetic diameters of these seven molecules, sorted from smallest to largest, were He (2.6 Å) < H_2_ (2.89 Å) < CO_2_ (3.3 Å) < O_2_ (3.46 Å) < H_2_S (3.62 Å) < N_2_ (3.64 Å) < CH_4_ (3.8 Å). As can be seen from Figure 2a,b, within the range of PLD < 3.7 Å, the diffusion coefficient of molecules increases with the increase in PLD, showing a high positive correlation. After that, as the pore size increases and all gases enter the interior of MOFs, when PLD > 5 Å, diffusivity decreases and tends to stabilize. Secondly, based on the diffusion coefficients of CH_4_ and N_2_ in the infinite dilution state, a quantitative relationship with PLD was established as log *D* = a PLD − b, as shown in Appendix A. We found that the molecules affected by PLD in descending order were CH_4_ > N_2_ > H_2_S > O_2_ > H_2_ > CO_2_ > He. Since CH_4_ has the largest molecular diameter among the four gases and He has the smallest, larger kinetic diameters slow gas molecule diffusion, enabling methane capture and gas separation, which is consistent with the conclusions reported previously in the literature [42].

We then analyzed the relationship between selectivity (x = N_2_, H_2_S, O_2_, CO_2_, H_2_, He) and PLD and LCD. Most MOFs with larger diffusion selectivity have relatively smaller PLD and LCD. From Figure 2c, it can be seen that when the PLD is between 2.4~4.2 Å, the adsorption selectivity for CO_2_/CH_4_ decreases. When PLD > 5 Å, both gas molecules can pass through freely, and the adsorption selectivity gradually stabilizes and tends towards 1. This is because CO_2_ has a strong quadrupole moment and is preferentially adsorbed over CH_4_ even in infinitely large pores, thus achieving the separation of CO_2_ from CH_4_. The pore size of MOFs can be precisely controlled, allowing only specific-sized molecules to pass through. Smaller pore sizes can prevent larger molecules from passing while allowing smaller molecules to diffuse freely. The molecular kinetic diameter of CO_2_ is about 3.3 Å, also smaller than that of the methane molecule. Therefore, designing MOFs with a pore size smaller than the kinetic diameter of the methane molecule can promote the permeation of CO_2_ molecules while hindering CH_4_ molecules, achieving CO_2_/CH_4_ separation. Similarly, the kinetic diameter of O_2_ is about 3.46 Å, and that of CH_4_ is about 3.8 Å. If the pore size of MOFs is designed to be smaller than 3.5 Å, it will be more conducive to the rapid diffusion of oxygen molecules and restrict the passage of methane molecules, achieving O_2_/CH_4_ separation. Other gases also show the same trend, as shown in Appendix A. Therefore, by finely tuning the pore size and channel geometry, a certain degree of methane purification can be achieved.

*S_diff_* decreases and then tends to flatten as the four descriptors (LCD, *ϕ*, VSA, PLD) increase, as shown in Figure 2. This is also due to the gradual release of strong steric hindrance. When the pore size is small, the pore walls of different framework materials have a strong or weak adsorption effect on gases, resulting in high selectivity differences. When *ϕ* is high, gas molecules have more space for free diffusion, so the diffusion capacity *S_diff_* is larger. When VSA is close to 0, that is, the specific surface area of MOFs is very small, and the pores are very limited, the diffusion area *S_diff_* of gas molecules is the largest. This is because at this time, MOF molecules either cannot pass any molecules or can only pass a small number of molecules other than methane. As VSA increases, that is, the specific surface area increases, the pores become more tortuous and complex, leading to a longer diffusion path for gas molecules, thereby reducing the diffusion capacity *S_diff_*. Finally, because further increasing VSA has a limited effect on increasing the diffusion path, or the pore structure is already complex enough that the diffusion capacity of gas molecules no longer decreases significantly, the diffusion selectivity equals 1, that is, methane purification cannot be achieved. The complexity of the pore structure of MOFs may have an important impact on the diffusion of gas molecules. As shown in Figure 2e,f, PLD also affects the diffusion of gas molecules. A smaller pore size may limit the diffusion of gas molecules, while a larger pore size may help to improve the diffusion rate. Other gases also show the same trend, as shown in Appendix A.

The results show that PLD is a key indicator of x/CH_4_ separation performance, but not a perfect one. The above univariate analysis can only preliminarily determine the relationship between a single parameter and performance. To screen structural variables that have a strong impact on all gas components and further predict the performance of MOF structures, we will further use ML algorithms for systematic multivariate analysis of the comprehensive structure–performance relationship of CoRE-MOFs.

### 3.2. Machine Learning

ML (machine learning) was used to model these complex large-scale data, and the prediction results were effective. Therefore, in this work, ensemble algorithms were employed in machine learning (ML) to enhance the performance of the predictive model, especially in forecasting diffusivity and diffusion selectivity. Details of *R*^2^, MAE, and RMSE are presented in Table 1. It can be seen that the computational accuracy of the RF algorithm is relatively low, and its stability is poor. Relatively speaking, this algorithm has lower predictive performance for methane purification. GBRT usually reduces prediction errors step by step in the iterative process of multiple decision trees, with each tree trying to correct the errors of the previous one. This strategy makes GBRT slightly better in prediction accuracy than RF.

Figure 3 shows that the prediction results of XGB and LGBM are very close, with considerable accuracy and stability. The *R*^2^ value of XGB is 0.957, and the *R*^2^ value of LGBM is 0.954; both are close to 1, indicating that both models fit the training data very well. The RMSE of XGB is 0.199, and the RMSE of LGBM is 0.207. Given the similar *R*^2^ and RMSE outcomes, we can deduce that both algorithms exhibit comparable accuracy.

Figure 4b shows the time performance of different machine learning algorithms in predicting diffusivity and selectivity, with the selectivity prediction time being shorter than that of diffusivity. Although the *R*^2^ value of the XGB algorithm is slightly larger than that of LGBM, the computing time of XGB is over 300 s. The CPU computation time of LGBM is significantly reduced; especially, when predicting selectivity, it reached an impressive speed of 23 s. Therefore, comparing the four algorithms comprehensively, it can be concluded that LGBM > XGB > GBRT > RF, with LGBM having the highest fitting ability, stability, and efficiency. The excellent predictive effect of LGBM may be attributed to its difference in tree structure from other tree model algorithms. Traditional tree models use a depth-first strategy when growing, that is, building one branch at a time. LGBM adopts leaf-wise growth, which can lead to the asymmetry of the tree but usually finds better split points more quickly. The tree structure built from leaf growth may better reflect the interaction between MOF description and performance than trees grown horizontally. Another optimization of LGBM is histogram subtraction acceleration. With this method, after constructing a histogram of a leaf, the histogram of its sibling leaf can be obtained at a very low cost, which can double the speed. Therefore, LGBM is the fastest among the four algorithms.

To avoid the impact of the training sample on the coverage of molecules in the test set, we calculated each of the six systems separately, as shown in Appendix A. As the training set sample size increases, the model is exposed to more data and can better learn the underlying patterns. Consequently, its predictive accuracy and generalization ability typically improve. Ideally, the improvement in accuracy observed on the training set should also be reflected in an independent test set, indicating that the model not only performs well on the training data but also generalizes effectively. For datasets with a small amount of data, the training score of LGBM is higher than the test score. Providing more data for training will enhance the model’s generalization ability and reduce overfitting. Surprisingly, the accuracy of the LGBM model for predicting small sample sizes is still very high (*R*^2^ = 0.910). Therefore, when the algorithm focuses on predicting within a certain range (for example, parts of materials with excellent diffusivity performance), the LGBM model still has a high degree of accuracy and can also be applied to predict the diffusivity performance of new gases or new MOF materials.

### 3.3. SHAP Analysis

To further explore the impact of MOF descriptors on diffusion performance, in this work, SHAP was utilized to interpret their relationships. Calculating SHAP values aims to quantify the impact of features on performance in terms of magnitude (significant or insignificant) and direction (positive or negative correlation). As shown in Figure 5a, the SHAP values based on the LGBM algorithm illustrate the impact on diffusivity. The influence is ranked as PLD > *pol* > *Dia* > VSA > LCD > LCD/PLD > *ϕ* > *ρ* > *Qua* > *Dip*. PLD is positively correlated with gas diffusion performance, meaning that an increase in PLD usually leads to an improvement in diffusion performance. The size of PLD can determine the MOF’s screening ability for gas molecules of different sizes, allowing smaller gas molecules to pass through quickly while hindering the diffusion of larger molecules, and achieving size-selective separation. A larger PLD provides more space for gas molecules to enter and diffuse, reducing intermolecular collisions and the diffusion resistance within the adsorbent, possessing a more open pore structure, and providing more diffusion channels for gas molecules. Therefore, the degree of match between the gas molecule’s kinetic diameter and PLD is an important factor in determining diffusion performance. This is consistent with previous research results [43], further verifying the importance of PLD in MOF design. *Pol*, as the second most important influencing factor, shows a significant negative correlation, meaning that an increase in the *pol* descriptor will reduce gas diffusion performance. *Dia*, the descriptor for gas molecules, mostly has a negative impact on gas diffusion, but there are also some positive correlations. This may indicate that the impact of *Dia* on diffusion performance depends on the specific MOF structure and other factors. An increase in VSA and LCD is found to promote gas diffusion. This may be because a larger VSA and LCD provide more adsorption sites and more open diffusion channels. Different MOF descriptors may interact with each other, and they collectively determine the MOF’s diffusion capability. By identifying and quantifying the impact of these descriptors, strategies can be provided for the optimization of MOF materials, such as adjusting pore size and surface chemical properties to improve the diffusion performance of target gases.

As shown in Figure 5b, the SHAP values based on the LGBM algorithm illustrate the impact on diffusion selectivity. The influence is ranked as *Dia* > *pol* > PLD > VSA > *ρ* > *ϕ* > LCD > Δ*Pol* > LCD/PLD > *Qua* > Δ*Dia >* Δ*Qua >* Δ*Dip > Dip*. *Dia* (as the most significant influencing factor)’s increase will have a negative correlation effect on diffusion selectivity, meaning that a larger *Dia* value will reduce the MOF’s diffusion selectivity, which has been mentioned in previous studies [44]. *Pol*, as the second influencing factor, also inhibits diffusion selectivity, indicating that stronger intermolecular forces will reduce the MOF’s separation efficiency. These two factors are the physical properties of the gas molecules themselves, indicating that the physical properties of the gas molecules have a significant negative impact on MOF diffusion selectivity [45]. In addition, PLD, VSA, and *ρ* each have an inhibitory effect on diffusion selectivity to varying degrees. An increase in PLD inhibits diffusion selectivity, possibly because an overly large PLD allows non-target molecules to pass through, reducing selectivity. An increase in VSA and *ρ* leads to a longer diffusion path for gas molecules within the MOF, increasing the resistance to mass transfer. Meanwhile, an increase in *ϕ* and LCD is positively correlated with diffusion selectivity, indicating that optimizing these parameters can improve the MOF’s diffusion selectivity.

### 3.4. Top-Performing Metal–Organic Frameworks (MOFs)

Further research has found that, when the LCD value is smaller, the LCD/PLD ratio is lower, the channels appear more uniform, and the diffusion performance is better. When the LCD/PLD ratio is around 1, the diffusion rate rises sharply, reaching a peak in diffusion. As the LCD/PLD value increases, the diffusion rate shows a downward trend, indicating that more uniform channels are conducive to diffusion. For low PLD (below 5 Å) with smaller LCD, *ϕ*, and VSA, the SHAP value is greater than 0, having a positive impact. When the MOFs’ PLD is between 2.4 and 5.0 Å, LCD is within 5 Å, *ϕ* is near 0.1, and the specific surface area is around 500, there will be a peak with a SHAP value greater than 0, which will have a positive effect on the target values of our ML model. Therefore, MOFs with higher diffusion selectivity can be obtained within this range.

To select CoRE-MOFs with excellent performance, we set limiting conditions according to the above range, as shown in Appendix A. In the selection process, *D*_i_ and *S_diff(i/j)_* are the two factors we jointly consider. As can be seen from Appendix A, the specific strategy involves screening for MOF materials that exhibit high diffusivity and significant selectivity for the target gas *i* to be separated. In Table 2, three optimal MOFs are selected for each system, and most MOFs have a larger *S_diff(i/j)_*. Among the materials selected in the He/CH_4_ system, ELUQIM04 has the largest diffusion selectivity, *S_diff(He/CH_*_4*)*_ = 61,406.36. Among the materials selected in the H_2_/CH_4_ system, ELUQIM05 has the largest diffusion selectivity, S*_diff(H_*_2/*CH*4*)*_ = 40,173.07. Figure 6a,b are the structural diagrams of the optimal MOFs, and Figure 6c,d are their corresponding pore diagrams. It can be seen that their channel structures are all circular, and it is speculated that the circular channels make the gas diffusion more uniform. In the selection of the six optimal MOFs for the system, the ranges of PLD and LCD are very concentrated, with PLD between 2.40 and 3.25 Å and LCD concentrated between 2.47 and 3.84 Å. This is because, within this range of PLD, CH_4_ cannot enter the channels due to its large molecular kinetic diameter, which enhances the separation performance of the six gases. It can be seen from the SHAP relative importance analysis that PLD is an important descriptor affecting gas separation. PLD close to the kinetic diameter of the separated gas is the key condition to achieve CH_4_ purification with good separation effects. This will provide effective theoretical guidance for the design of new MOFs with excellent performance.

### 3.5. Design Strategies of MOFs with High Performances

To further guide experiments and design novel MOF materials with excellent methane purification and diffusion selectivity, this section discusses a series of design strategies for hypothetical MOF materials. Each pair comprises one MOF with superior diffusion selectivity and another with inferior performance, differentiated by a single compositional variation, such as the metal center, organic linker, or topology. Details of MOF composition are shown in Table 3. Based on this, we propose three strategies to diffuse gases from the six systems to achieve efficient methane purification, similar to previous studies [46].

In Table 3a, the alteration of the organic linker in this group of MOFs resulted in a dramatic increase in selectivity, jumping from 7.93 to 643.48—a remarkable enhancement of 80.15 times. As shown in Table 3e, in this group of MOFs, the sole modification was the metal center, with the higher performing MOF featuring Zn. Zn’s atomic radius of 0.142 nm was more compact than Cd (0.151 nm), a difference that influenced the pore structure and dimensions of the material. Additionally, metal activity significantly influenced adsorption, with Zn noted for its greater reactivity compared with Cd, the selectivity was improved from 1.22 to 21.93, an increase of 16.97 times. In Figure 7c, a modification was made solely to the MOF’s topology. By substituting the fsc topology with pst, the selectivity saw a remarkable enhancement, soaring from 5.36 to 147.93—a 26.6-fold improvement in performance. Variations in separation efficacy stem from the distinct pore dimensions and shapes created by the MOFs’ topological networks. Ideally, in gas separation processes, pore sizes are meticulously regulated to permit the passage of molecules of a particular size while excluding others, ensuring selective gas permeation. It is evident that by adjusting topologies, organic linkers, or metal centers, one can obtain or design superior MOF adsorbents tailored for various systems.

## 4. Conclusions

In this study, big data and machine learning were combined to simulate the diffusion and selectivity for methane purification in 6013 CoRE-MOFs within six gas mixtures (H_2_/CH_4_, N_2_/CH_4_, O_2_/CH_4_, CO_2_/CH_4_, H_2_S/CH_4_, He/CH_4_). Preliminary analysis indicated a close relationship between *D* and PLD, establishing a quantitative relationship of log *D* = a PLD − b. Subsequently, four machine learning algorithms were applied to predict the diffusion and selectivity of MOFs in the six systems. The result analysis showed significant performance differences among the models, ranked by excellence as LGBM > XGB > GBRT > RF. Further SHAP relative importance analysis using the LGBM algorithm indicated that PLD is the most important structural descriptor affecting diffusivity, and *Dia* is the most important descriptor affecting selectivity. Additionally, three optimal MOFs were selected for each of the six systems, further verifying that the diffusion properties of gases are key to separating other gases from methane for its purification. Finally, three design strategies were proposed to effectively purify methane for specific systems by replacing different metal nodes, adjusting various topologies, and changing multiple organic linkers. This research provides effective guidance for experimentalists to select high-performance MOFs to separate other gases from methane.

## Figures and Tables

**Figure 1 nanomaterials-14-01074-f001:**
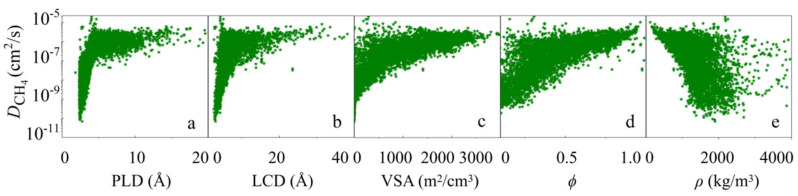
The variation of CH_4_ diffusion performance with (**a**) PLD, (**b**) LCD, (**c**) VSA, (**d**) *ϕ,* (**e**) *ρ*.

**Figure 2 nanomaterials-14-01074-f002:**
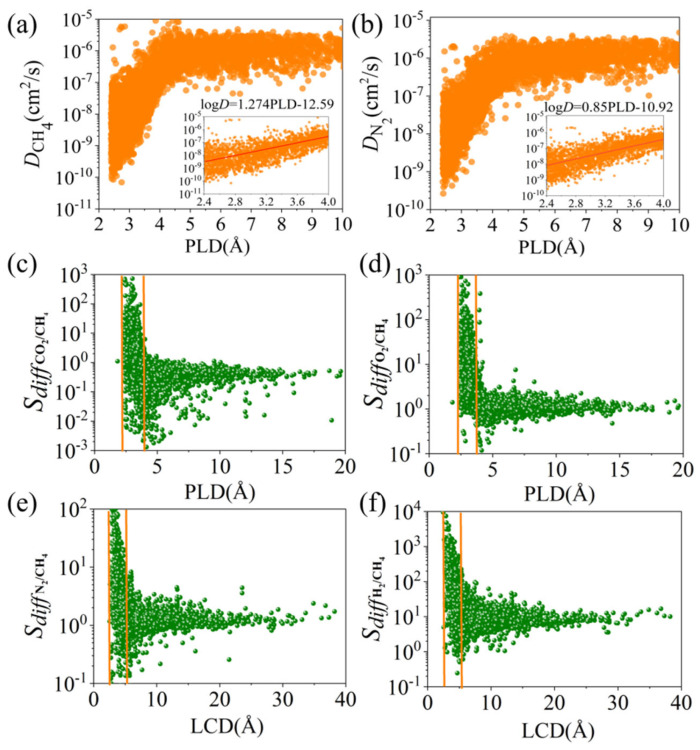
The effect of MOFs’ PLD and LCD on the diffusion properties of gases for (**a**) CH_4_—PLD, (**b**) N_2_—PLD, (**c**) CO_2_/CH_4_—PLD, (**d**) O_2_/CH_4_—PLD, (**e**) N_2_/CH_4_—LCD, and (**f**) H_2_/CH_4_—LCD. (The orange line represents the range of optimal MOFs).

**Figure 3 nanomaterials-14-01074-f003:**
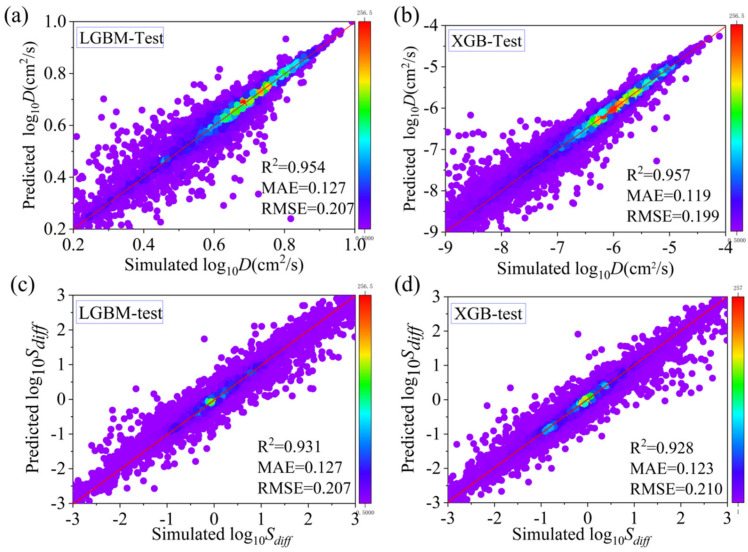
Prediction of machine learning for diffusivity and diffusion selectivity with (**a**) the predictions of *D* by LGBM, (**b**) the predictions of *D* by XGB, (**c**) the predictions of *S_diff_* by LGBM, and (**d**) the predictions of *S_diff_* by XGB. (The color of the dots represents the quantity).

**Figure 4 nanomaterials-14-01074-f004:**
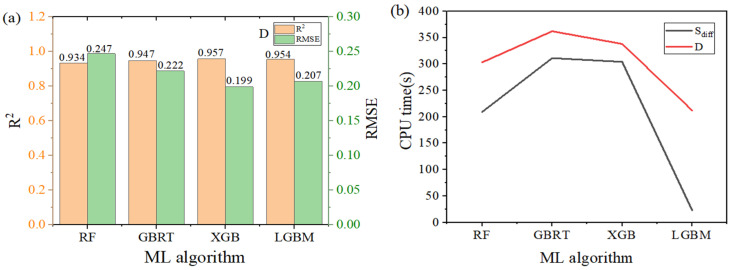
Comparison of the four algorithms. (**a**) *R*^2^ and RMSE, (**b**) *D* and *S*.

**Figure 5 nanomaterials-14-01074-f005:**
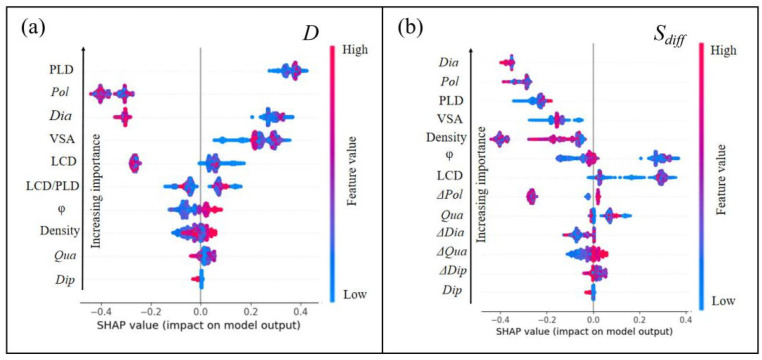
Relative importance analysis chart of structural descriptors for (**a**) *D* and (**b**) *S_diff_*.

**Figure 6 nanomaterials-14-01074-f006:**
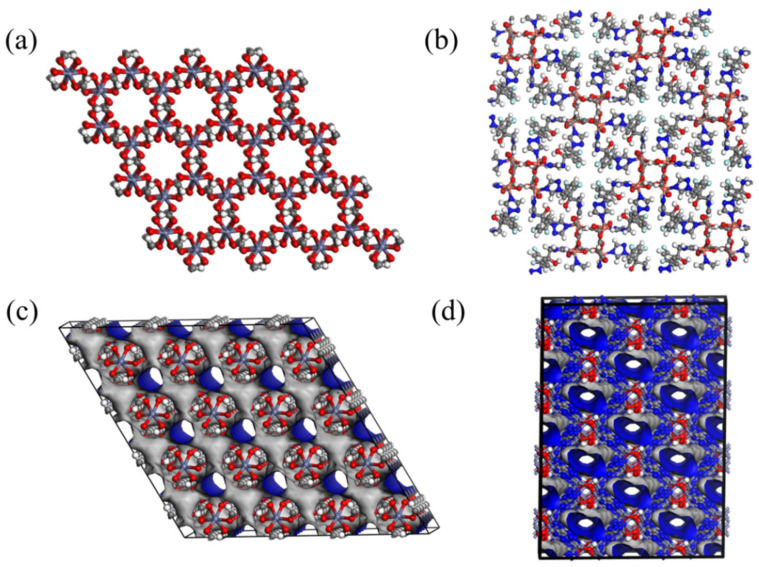
The structure of MOF for (**a**) ELUQIM04 and (**b**) GUXQAS; the pore channel diagram of MOF for (**c**) ELUQIM04 and (**d**) GUXQAS. (Color code: O atoms of MOFs: red; C atoms: gray; H atoms: white; N atoms: blue).

**Figure 7 nanomaterials-14-01074-f007:**
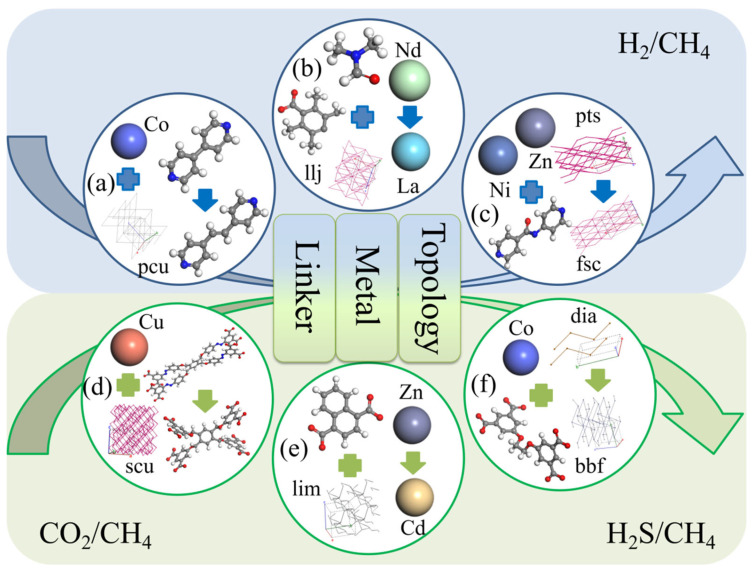
Enhancement of gas diffusion selectivity performance by three different MOF design strategies. The different pairs of MOFs for (**a**) GIRDUI and GIRGUL; (**b**) CIMTAV and CIMTEZ; (**c**) EBELUU and EBEMEF; (**d**) CAJQEL and CAJQIP; (**e**) HEBTEP and SETFUT; (**f**) ISIKIF and ISIKOL.

**Table 1 nanomaterials-14-01074-t001:** Comparative prediction of *R*^2^, MAE, and RMSE for *D* and *S* performance by four machine learning algorithms.

Indicators	Algorithm	*R* ^2^	MAE	RMSE
*D*	RF	0.934	0.151	0.247
	LGBM	0.954	0.127	0.207
	XGB	0.957	0.119	0.199
	GBRT	0.947	0.129	0.222
*S*	RF	0.887	0.155	0.264
	LGBM	0.931	0.127	0.207
	XGB	0.928	0.123	0.210
	GBRT	0.907	0.135	0.240

**Table 2 nanomaterials-14-01074-t002:** The best-performing MOFs selected from six systems.

Gas Mixture *i/j*	CSD Cord	LCD [Å]	*ϕ*	PLD [Å]	*ρ* [kg/m^3^]	*D_i_* [cm^2^/s]	*D_j_* [cm^2^/s]	*S_diff(i/j)_*
He/CH_4_	ELUQIM04	2.92	0.04	2.44	1764.65	9.03 × 10^−6^	1.47 × 10^−10^	61,406.36
	ELUQIM05	2.90	0.04	2.43	1773.83	8.42 × 10^−6^	1.76 × 10^−10^	47,815.36
	ELUQIM06	2.89	0.04	2.41	1779.43	9.74 × 10^−6^	8.11 × 10^−10^	12,015.02
H_2_/CH_4_	ELUQIM05	2.90	0.04	2.43	1773.83	7.08 × 10^−6^	1.76 × 10^−10^	40,173.07
	ELUQIM04	2.92	0.04	2.44	1764.65	4.14 × 10^−6^	1.47 × 10^−10^	28,148.27
	FAPYEA04	2.47	0.00	2.40	1583.54	2.83 × 10^−6^	3.01 × 10^−10^	9397.11
CO_2_/CH_4_	XEKDUO	2.98	0.02	2.75	1903.55	1.70 × 10^−7^	6.92 × 10^−11^	2463.01
	ELUQIM05	2.90	0.04	2.43	1773.83	3.91 × 10^−7^	1.76 × 10^−10^	2220.35
	HIQPEE	3.84	0.15	3.12	1440.14	1.24 × 10^−6^	5.69 × 10^−10^	2185.18
O_2_/CH_4_	FAPYEA04	2.47	0.00	2.40	1583.54	2.92 × 10^−7^	3.01 × 10^−10^	968.16
	ELUQIM05	2.90	0.04	2.43	1773.83	1.58 × 10^−7^	1.76 × 10^−10^	898.78
	GUXQAS	2.79	0.02	2.52	1598.28	1.16 × 10^−7^	1.30 × 10^−10^	893.53
H_2_S/CH_4_	GUXQAS	2.79	0.02	2.52	1598.28	1.70 × 10^−9^	1.30 × 10^−10^	13.04
	RUPZIM	3.48	0.11	3.25	1549.49	1.51 × 10^−7^	1.39 × 10^−8^	10.87
	GUXPUL	2.79	0.02	2.58	1595.02	1.18 × 10^−9^	1.10 × 10^−10^	10.70
N_2_/CH_4_	FAPYEA04	2.47	0.00	2.40	1583.54	1.11 × 10^−7^	3.01 × 10^−10^	369.05
	PARFOF	2.77	0.05	2.46	1541.02	4.78 × 10^−7^	2.16 × 10^−9^	221.18
	HIWXER01	3.29	0.13	2.76	2533.00	1.80 × 10^−7^	1.27 × 10^−9^	141.34

**Table 3 nanomaterials-14-01074-t003:** Comparison of the structures of different pairs of MOFs.

Gas Mixture (i/j)	NO.	CSD Cord	Metal Center	Organic Links	Top Structure	*S_diff(i/j)_*
H_2_/CH_4_	a	GIRDUI	Co	MGFJDEHFNMWYBD	pcu	7.93
		GIRGUL	Co	MTAVBTGOXNGCJR	pcu	643.48
	b	CIMTAV	La	BVKZGUZCCUSVTD	llj	4.58
		CIMTEZ	Nd	BVKZGUZCCUSVTD	llj	12.97
	c	EBELUU	NiZn	JEVCWSUVFOYBFI	fsc	5.36
		EBEMEF	NiZn	JEVCWSUVFOYBFI	pts	147.93
CO_2_/CH_4_	d	CAJQEL	Cu	GRYHAGOZZMMYAO	scu	0.31
		CAJQIP	Cu	DUKMDOUQAIDJRW	scu	2.24
	e	HEBTEP	Zn	ABMFBCRYHDZLRD	lim	21.93
		SETFUT	Cd	ABMFBCRYHDZLRD	lim	1.22
H_2_S/CH_4_	f	ISIKIF	Co	GEBVRXNOWAYDCP	dia	16.07
		ISIKOL	Co	GEBVRXNOWAYDCP	bbf	0.97

## Data Availability

Data are contained within the article and Appendix A.

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
