# Peer review of "Data-Driven and Machine Learning to Screen Metal–Organic Frameworks for the Efficient Separation of Methane"

_nanomaterials, 2024, doi:10.3390/nano14131074_

Round 1

Reviewer 1 Report

Comments and Suggestions for Authors

The work presents high-throughput computational screening of MOFs for the diffusivities of various gases, forming the basis for CH4-relevant separations. The results are analyzed to provide an initial assessment of correlations between structural MOF characteristics and gas penetrant characteristics with diffusivities and selectivity. Subsequently, the authors employ various ML models for diffusivity and selectivity prediction, ultimately selecting the best-performing model. This is followed from a more thorough feature-importance analysis with SHAP. Interestingly, the authors propose modifications to improve the performance of existing MOFs by replacing the metal and the linker and changing the topology of the framework.

My issue is with the last section (Design Strategies of MOFs with High Performances): Firstly, I found it very challenging to follow Figure 7 and the entire modification process related to it. This section needs to be re-written for clarity. Additionally, a table showing the changes in selectivity performance would help readers better understand the improvements from the modifications. Moreover, it is unclear how the authors utilized the knowledge gained from the SHAP analysis to determine the specific modifications (linker, metal, topology) for the selected cases.

Furthermore, there is a trade-off between the diffusivity of the fast-diffusing species, i, of a pair i and j, and the selectivity (Di/Dj). By improving selectivity, there might be a drop in Di that makes the separation process less viable for industrial applications. The goal of such a modification strategy should be to improve Di/Dj without significantly compromising Di. Including a discussion on this trade-off would enhance the manuscript.

Some additional minor issues:
- The first citation is numbered 0 instead of 1.

- In the paragraph before Section 3.3 the authors write "To avoid the impact of the training sample on the coverage of molecules in the test set, we calculated each of the six systems separately, as shown in Table S5." However, Table S5 shows the packages version

Reviewer 2 Report

Comments and Suggestions for Authors

The authors present a nice, neat paper with the key result showing the performance of four common machine learning algorithms for the prediction of gas separation in MOFs. This is a useful result to publish and will help guide other researchers.

That said, there are a few aspects of the current manuscript that are slightly mis-targeted. A large part of the introduction and (especially) methods is written more in the style of an undergrduate text or tutorial review (e.g. the description of PBC on page 7, MAE etc on page 9).

Some more specific queries:

Page 6: How were the parameters listed in Table S2 derived? This requires explanation

Page 7: "..simulation using this method showed better consistency..." better than what? This needs to be described and shown.

Page 8: What does it mean in this context to arrange data longitudinally?

Page 11: "Since CH4 has the largest kinetic diameter ... molecules with smaller kinetic diameters diffuse faster..." This doesn't follow - the second half onf the sentence has no direct relationship to methane. It is also rather obvious.

Page 13: The MOFs are periodic structures, not molecules. Just using "MOFs" or "MOF structures" is fine.

Page 15: What is the "test" indicated in the key of each plot in Figure 3?

SI Table S10: I think "pairs" of MOFs is intended, not "parts"

Comments on the Quality of English Language

My mark for the quality of English takes into account the overall "tutorial" feel of the manuscript. The English itself is fine, but a paper that only uses common methods (distinct from developing or modifying them) doesn't need to explain each component so much.
